# A Cross-Sectional Study of Veterinarians in Germany on the Impact of the TÄHAV Amendment 2018 on Antimicrobial Use and Development of Antimicrobial Resistance in Dogs and Cats

**DOI:** 10.3390/antibiotics11040484

**Published:** 2022-04-05

**Authors:** Marianne Moerer, Roswitha Merle, Wolfgang Bäumer

**Affiliations:** 1Institute of Pharmacology and Toxicology, Department of Veterinary Medicine, Freie Universität Berlin, Koserstraße 20, 14195 Berlin, Germany; marianne.moerer@fu-berlin.de; 2Institute for Veterinary Epidemiology and Biostatistics, Department of Veterinary Medicine, Freie Universität Berlin, Königsweg 67, 14163 Berlin, Germany; epi@vetmed.fu-berlin.de

**Keywords:** antibiotic, AST, companion animals, HPCIA, survey

## Abstract

To minimize the use of third- and fourth-generation cephalosporins and fluoroquinolones, the 2018 amendment to the regulations of veterinary pharmacies (TÄHAV) introduced legal restrictions in Germany. In an online survey among German veterinarians, we investigated the influence of these requirements on the use of antibiotics in the treatment of dogs and cats and the development of resistance rates. It was found that, on average, between 21% and 30% of daily treated dogs and cats received antimicrobial therapy. The TÄHAV amendment led to a less frequent use of highest priority critically important antimicrobials (HPCIA) in 79% (240/303) of respondents and less antimicrobial use in general in 36% (108/303). As a result of these legal changes, 63% (190/303) of participants requested antimicrobial susceptibility testing (AST) more frequently. Participants consulted ASTs particularly frequently for treatment of otitis externa with 63% (190/303), cystitis with 55% (168/303), wounds with 44% (132/303), and pyoderma with 29% (88/303). Veterinarians also noted an increased loss of antimicrobial efficacy, especially when treating these diseases. The results of our survey confirm that the TÄHAV amendment is having a positive impact on prudent antibiotic use, with participants performing more ASTs, using HPCIA less frequently, and choosing alternative antimicrobials for therapy.

## 1. Introduction

Antimicrobial resistance is a key reason for treatment failure of bacterial infections [1] and has therefore been considered one of the most relevant issues in public health. Use of antimicrobials, especially their misuse, is recognized as one of the main drivers for the selection of antibiotic-resistant bacteria [2,3,4,5,6]. Responsible antibiotic application and rational use is therefore considered one of the most important strategies in combatting antimicrobial resistance in both human and veterinary medicine [7].

Due to a big overlap of antimicrobials used in companion animals and human beings [8], the threat for human health posed by antimicrobial use in animals arises from the possibility of selecting resistant strains and transmission of resistance through the environment and close contact between humans and their pets [9,10,11,12]. Therefore, an increasing number of guidelines restrict the use of “highest priority critically important antimicrobials” (HPCIA), a term coined by the World Health Organization (WHO). In particular, third- and fourth-generation cephalosporins and fluoroquinolones are being regulated, since these are without an alternative for the treatment of certain human diseases [13]. 

As a measure to reduce resistance, the German veterinary profession established guidelines for the responsible use of antimicrobials in 2000 [2]. With the antibiotic minimization concept in food-producing animals, which was introduced in 2014 with the 16th amendment of the Medical Products Act (Arzneimittelgesetz), there has already been a reduction in total antibiotic use in food-producing animals by more than 50% [14]. 

As German veterinarians have the right to prescribe and dispense medicines, there is a specific legislation regulating the details of dispensing medicines (Verordnung über tierärztliche Hausapotheken (TÄHAV)). The aim of the 2018 amendment to the TÄHAV is to mitigate the development and spread of resistance by minimizing the number of antibiotic treatments to the therapeutically necessary level in order to maintain the effectiveness of antibiotics, especially of third- or fourth-generation cephalosporins and fluoroquinolones. Therefore, in addition to clinical examination, susceptibility testing of pathogenic bacteria to the available antimicrobial agents is concretized as an important element for therapy decision. Veterinarians are now required to perform antimicrobial susceptibility testing (AST) for the use of third- or fourth-generation cephalosporins and fluoroquinolones, even for the treatment of individual animals, such as dogs and cats. However, the rule of performing AST can be disregarded if the health of the animal is endangered by the sampling, the pathogen cannot be cultured in a cell-free medium, or a suitable method for determining the sensitivity of the pathogen is not available [15].

Since the amendment has been in force for about three years, the research question is whether these legal changes have led to an adaption concerning antimicrobial prescription behavior and AST practice [16] and whether a positive impact on the resistance development might have been achieved through this legal amendment. 

This study aims to explore the influence of the TÄHAV amendment 2018 on the prescription patterns of antimicrobials as well as susceptibility testing practice as perceived by German veterinarians through a survey. The development of resistance is estimated by our participants. 

## 2. Results

The survey was answered by 378 participants. A total of 303 questionnaires were used for the evaluation because a minimum of information was reached (19 questions concerning practice information and general information on antimicrobial use). This corresponds to 3% of registered small animal veterinarians in Germany according to an analysis of the Federal Veterinary Association (Bundestierärztekammer) in 2020 [17].

### 2.1. Practice Information

While 72% (217/303) reported to be practice owners, 28% (85/303) claimed to be employed veterinarians. A total of 93% (281/303) of participants worked in a small animal practice, and 7% (22/303) in a clinic. In Germany, a veterinary clinic must ensure constant availability for service. A total of 58% (176/303) of respondents worked in a small town (fewer than 20,000 residents), 26% (80/303) reported being in a medium-sized city (20,000–100,000 residents), and 16% (47/303) stated they work in a large city (more than 100,000 residents).

### 2.2. Antibiotic Use

Participants were asked to estimate their daily use of antimicrobials. On average, antimicrobial use was found to range between 21% and 30% (Figure 1). 

A trend toward increased antibiotic use was found for participants with older age than for younger respondents during this study (*p* = 0.087, linear regression). Moreover, participants with infrequent antibiotic use were found to report lack of efficacy for antibiotic treatment of pyoderma (*p* = 0.022) and cystitis (*p* = 0.003) significantly less often.

The following responses are presented in Table 1. Penicillins were described as the most frequently used group of agents as reported by 93% (283/303) of our participants. In contrast, HPCIA were frequently used by only 6% (18/303). Seventy nine percent (240/303) of respondents indicated a less frequent use of HPCIA since the 2018 amendment to the TÄHAV went into effect. In 36% (108/303) of cases, veterinarians felt the amendment had also led to less antimicrobial use in general. Since the implementation of the TÄHAV amendment, 63% (190/303) of participants perceived AST as requested more frequently, but 17% (53/303) of respondents reported that their testing behavior had not changed. However, 76% (230/303) of participants experienced penicillins being used generally as a first-line treatment because of this amendment. In this context, 24% (73/303) of the respondents noticed penicillins had to be changed particularly frequently after AST. A total of 45% (135/303) of participating veterinarians stated they requested AST more frequently alongside treatment, and 20% (62/303) waited for the test result to choose the antimicrobial agent. ASTs were perceived as particularly often consulted for the treatment of otitis externa (63%, 190/303), cystitis (55%, 168/303), wounds (44%, 132/303), pyoderma (29%, 88/303), respiratory infections (23%, 71/303), and diarrhea (16%, 47/303). However, only 27% (77/289) of respondents reported testing frequently or always for cystitis therapy, as well as 13% for otitis externa (39/300) and pyoderma (39/292) therapy, and 8% (24/290) for bite wounds (Table 2). Furthermore, participants observed poor efficacy of antibiotic drugs during treatment of cystitis (24%, 70/289) (Table 3, otitis externa (20%, 60/300) (Table 4), and pyoderma (13%, 39/292) (Table 5). Participants who noted a lack of efficacy observed this especially in active ingredients of the penicillin group (Cystitis 83%, 58/70 (Table 3); Pyoderma 87%, 34/39 Table 5). Interestingly, a lack of antimicrobial efficacy was significantly less frequently reported by participants with infrequent use of AST (*p* < 0.001).

When asked about the owners’ reactions to additional costs associated with AST, about half of the veterinarians (53%, 159/303) said that their patient owners were fine with additional costs. However, 47% (144/303) stated that only some patients’ owners agreed to AST. 

The questionnaire can be found in the Appendix A; the distribution of responses is shown in Table 1 for general questions about antibiotic use and Table 2, Table 3, Table 4, Table 5 and Table 6 for the specifically discussed diseases of cystitis, otitis externa, pyoderma, and bite wounds.

A cluster analysis revealed that our participants can be divided into three groups based on their prescription patterns. One group includes veterinarians that applied the rules of prudent use widely, described antibiotics use as infrequent, claimed antibiograms are performed frequently, and reported being influenced by the TÄHAV amendment in their use of antibiotics and number of tests. This group consisted of 26% of our participants (73/303) and was found to be more likely young, employed practitioners working in clinics in large cities. Veterinarians who described antibiotics as being used more frequently, claimed infrequent resistance testing, and were not influenced by the TÄHAV amendment were more likely to be older owners of smaller rural practices (second group, 59%, 170/303). The third group was characterized by veterinarians who felt using antibiotics frequently but also claimed performing AST frequently and frequently observing a lack of efficacy of antimicrobial agents (15%, 44/303). This group consisted of veterinarians from specialized practices that work more frequently with pretreated and referred patients. 

## 3. Discussion

The hypothesis of the present study was that the amendment of the TÄHAV 2018 has led to a more prudent antimicrobial use, especially of HPCIA, in German veterinarians treating companion animals. The prudent use of antimicrobials is widely recognized as an important strategy in combatting antimicrobial resistance [7,18] and is one of the five objectives of the WHO global action plan [19]. Although some authors associate rather the quality of antibiotic use and less the quantity with the development of resistance [5,20,21], selective pressure imposed by any use of antibiotics mainly affects the emergence of resistant bacteria [21]. Participants in our survey estimated antimicrobial use on average between 21% and 30% of their daily treated dogs and cats. These data are only subjective estimates; therefore, they cannot be considered reliable. A precise query of the quantities will only be possible from 2026, as the amendment to the Medicines Act will then oblige German veterinarians to report used antimicrobials. However, they correlate well with published data describing between 16% and 30% of companion animals being treated with an antibiotic [20,22,23,24,25].

Among small animal veterinarians participating in the current survey, different behavioral patterns in the use of antibiotics and a different perception of the development of resistance were revealed. Although this characterization may not apply to every physician, it was found that young veterinarians in large cities apply the rules of prudent use more often than rural veterinarians with many years of experience. This discrepancy could be associated with differences in the undergraduate training received and previous work experience [26]. 

Studies frequently present penicillins as the most commonly used antimicrobial group [16,20,23,24,26,27], which was also confirmed by the participants of the current survey. Aminopenicillins in particular are also often recommended in the literature as first-line therapy [28,29]. Due to their broad spectrum of activity, ease of administration, and good tolerability, this group of penicillins is increasingly used [4,25,30,31]. Some authors believe that the high use of penicillins represents an infrequent use of bacterial culture and AST by clinicians [24,32,33]. This statement is confirmed by the participating veterinarians, who said that due to the amendment of the TÄHAV, penicillins were increasingly being applied, as their use does not require resistance testing. Despite the frequent use of penicillins since the TÄHAV amendment, our participants have not noticed any increased loss of efficacy in medication with these active ingredients, and study data have not shown any increased rates of resistance to penicillins in veterinary medicine in Germany since 2018 [34]. However, older studies have already described increased levels of resistance to penicillin in Europe [4,35].

The WHO has compiled a list of antibiotic agents identified as HPCIA to maintain their efficacy for human medicine. Of this HPCIA group, fluoroquinolones and third-generation cephalosporins are frequently used in small animal medicine [22,36]. A wide variation in data between countries can be found in the literature. In this context, HPCIA accounted for less than 10% of the total antibiotic use in a study from Norway [31] and Germany [16], which can be confirmed by our results. However, in the UK, a high use of HPCIA, especially in cats, has been found [36]. Our results indicate that the requirements of the TÄHAV amendment result in a less frequent use of HPCIA. The implementation of antimicrobial policies at a clinic level is also reported in the literature as a positive impact on the use of HPCIA [18,20,37]. In addition to the reduced use of HPCIA, the TÄHAV amendment has also led our participants to describe AST as being increasingly used. It requires an AST prior to any use of a fluoroquinolone or cephalosporin of the third and fourth generation in dogs and cats, since AST is considered one of the most important factors regulating the selection of antimicrobials for clinical veterinary use [38]. 

AST incurs additional costs in the region of EUR 60 to EUR 70 for pet owners, which are not trivial. Surveys throughout Europe as well as our study found that the willingness and ability of pet owners to pay more influence the treatment decision greatly [18,25,26]. Therefore, it is very important to educate pet owners about the resistance development, since costs should not compromise good veterinary practice.

According to the literature, AST was most frequently performed for the treatment of deep pyoderma, otitis, wound infections, and urinary tract infections [7,35,39], which also correlates well with data of the study at hand. These diseases, along with respiratory infections and digestive disorders, are also the most common reasons for antibiotic therapy [4,30,33] and are frequently associated with bacterial species listed by the WHO as increasingly resistant germs [7,29,40]. Therefore, it is strongly recommended to perform AST on a regular basis in order to improve effective treatment [2,41]. Interestingly, this is less well reflected by our participants’ opinion, who rarely considered AST necessary for successful therapy, as they hardly noticed any loss of efficacy of frequently used antibiotics. This opinion is expressed by the majority of veterinarians from various European countries [18,26,30]. Therefore, almost half of the participants claimed not to wait for the results of the resistance test to start antibiotic therapy, but only sending in the sample to meet the amendment of the TÄHAV [25]. However, given the participants’ more frequent testing resulted in increased levels of resistance as perceived, it could be assumed that the regulations enforced by the amendment of the TÄHAV lead to more AST and an increased awareness of resistance development.

The current study tested hypotheses based on a qualitative interview with Berlin veterinarians [25]. Even though Berlin has a certain heterogeneity, the survey was extended to the whole country because the environment, as well as the population and thus the patients’ owners, differ in age, nationality, financial means, and their relationship with the animal, and therefore might have different demands in terms of medical care for their pets. Consequently, veterinarians throughout Germany may have different attitudes toward the use of antibiotics and the TÄHAV restrictions compared to veterinarians in Berlin.

The hypotheses based on the Berlin interviews were mainly confirmed in the subsequent Germany-wide survey. It was found that the estimated antibiotic use in Berlin, at 20%, is somewhat lower than the nationwide use of 21% to 30%. A positive influence of the TÄHAV amendment on the prudent use of antibiotics was confirmed, as both in Berlin and in the whole of Germany, HPCIA are perceived as being used less frequently, alternative antimicrobial agents to these groups are consciously chosen, and participants claim to consult AST more frequently. However, all in all, AST is perceived as still relatively rare. In both surveys, the group of penicillins was described as most frequently lacking efficacy, but an increased loss of efficacy was not observed by participants. However, since these are only subjective opinions, a valid statement on the development of resistance will be made in a follow-up study based on AST analysis.

### Limitations

Since participation in the survey was voluntary, it can be assumed that veterinarians with a greater interest in the topic are overrepresented and veterinarians with a high workload were less likely to participate. By advertising the questionnaire via social media and an additional telephone invitation of randomly selected veterinarians, probably not all German small animal veterinarians were reached, so the actual opinions may be biased. 

The survey was taken by about 3% of small animal veterinarians in Germany. The distribution between participating practice owners and employed veterinarians, as well as veterinarians employed in practices and clinics, is almost identical to the information provided in the reports of the Federal Veterinary Association [17]. Since the sample of participants is similar to the population of small animal practitioners in Germany in its composition and structure of relevant characteristics, the results can be considered fairly representative. All results of this study are based on subjective opinions of the participants. Since the recording of the use of antibiotics is currently not mandatory in Germany, no objective data are available. Therefore, a considerable difference between the opinion of the veterinarians and actual use cannot be completely ruled out.

## 4. Materials and Methods

### 4.1. Questionnaire

The survey was a cross-sectional study to assess the antibiotic use and resistance development in small animal practices and was a quantitative online survey. The questionnaire was designed in German language and included 47 single-choice questions, 1 multiple-choice question, and 1 open question asking for comments at the end. The content was based on the answers of a previous qualitative survey of small animal veterinarians practicing in Berlin regarding the same topic [25]. During these interviews, detailed information on opinions, views, and attitudes was collected and interpretatively processed. This in-depth description of the topic allowed building hypotheses, which were tested in the study at hand. The larger number of cases enables a valid statistical evaluation to explain causal relationships [42]. Because this questionnaire was developed from a previous Berlin-wide survey, which had a high level of matching responses, the π-value (percentage of the characteristic in the population) was increased. Thus, with a confidence interval of 95% and a margin of error of 5%, the sample size was 301 participants. The first part of the questionnaire asked for information about the practice, followed by a section with general questions and concluded with specific questions. The general questions included general antibiotic use and the changes in the use and handling since the TÄHAV amendment in 2018. In the specific part, the treatment of selected diseases was evaluated in more detail; these included otitis externa, pyoderma, bite wounds, and cystitis.

### 4.2. Implementation

The survey was published via LimeSurvey version 3.15.9+190214 (©2006–2021 LimeSurvey GmbH, Hamburg, Germany) and was open from May to November 2021. All veterinarians practicing in Germany were welcome to participate in the survey. For this purpose, the questionnaire was promoted via social media. Additionally, 600 randomly selected veterinary practices were contacted by telephone. If permission was granted, the link to the questionnaire was sent via e-mail to the respective 510 practices. A total of 378 veterinarians practicing in Germany participated in our survey. However, 75 questionnaires had to be excluded from the evaluation because the minimum level of information was not reached (19 questions on practice information and general questions on antimicrobial use). The 303 questionnaires included in our study represented 3% of veterinarians treating companion animals registered in Germany according to the Federal Veterinary Association (Bundestierärztekammer) in 2020. 

### 4.3. Analysis

For the analysis, the collected data were transferred to Microsoft Excel^®^ 2018 (Microsoft Corporation, 2018, Microsoft Excel. Retrieved from https://office.microsoft.com/excel, accessed on 30 June 2021), which was used to rewrite the obtained data into a code. This coded data were transferred to IBM SPSS version 27 (IBM Corp. Released 2020. IBM SPSS Statistics for Windows, version 27.0. Armonk, NY, USA: IBM Corp) for statistical analysis. 

Categorical variables were described using counts, percentages, and 95% confidence interval (95% CI) calculated using jaik (http://www.jaik.de/js/bin_konf.htm, accessed on 13 December 2021). Response categories that did not reach the 5% limit were meaningfully combined. Continuous data were described using the mean and the corresponding 95% CI. Binary logistic regression was performed to estimate probability associations, and the model was tested by the omnibus test, classification table, and Nagelkerke’s R-square. A standardized residual of 2 and greater, and −2 and less was considered significant. *p*-values less than 0.05 were considered statistically significant.

A cluster analysis using Euclidian distance and the Ward method was performed to cluster participants according to their response behaviors. The dendrogram, as well as the analysis of Mojena, revealed an optimum of three clusters.

The study was approved by the Ethics Committee of Freie Universität Berlin (ZEA 2021-016, 3 September 2021).

## 5. Conclusions

The current study shows that German veterinarians are making great efforts to use antibiotics responsibly. The amendment of the TÄHAV has led to a decreased use of HPCIA and increased the frequency of AST. Therefore, based on the data obtained, it can be concluded that legislative regulation is effective in minimization of the use of HPCIA. Further research is necessary to assess the effect of these legal changes regarding the development of resistance. 

## Figures and Tables

**Figure 1 antibiotics-11-00484-f001:**
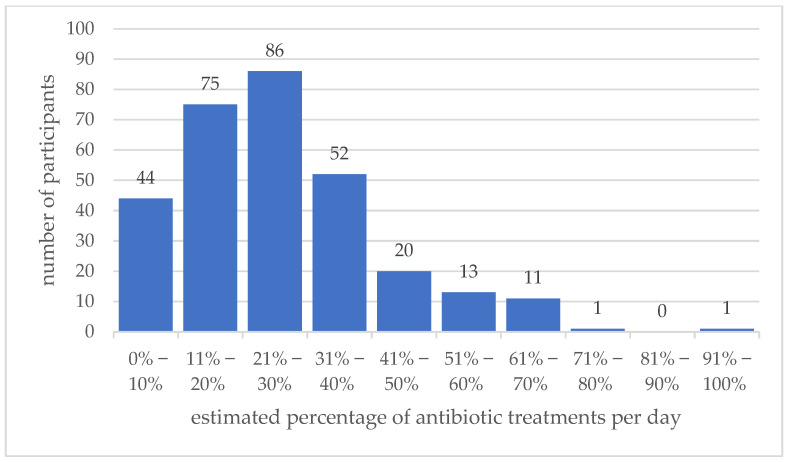
Antibiotic use. Number of participating veterinarians estimating how many dogs and cats treated daily (%) received antibiotic treatment. Bars indicate the percentages reported. On average, between 21% and 30% of cats and dogs treated daily received antimicrobial therapy. A total of 303 questionnaires were analyzed.

**Table 1 antibiotics-11-00484-t001:** Number, percentages, and 95% CI of responses to general antibiotic use as well as the impact of the TÄHAV amendment 2018 on the use of antimicrobials and AST. A total of 303 questionnaires of a survey among German veterinarians were analyzed.

	Absolute Number	Percentages	95% CI
Do you use topical or systemic antibiotics more frequently?			
Topical more frequent	19	6.3%	4.05–9.59%
Systemic more frequent	186	61.4%	55.79–66.69%
About equally often	98	32.3%	27.33–37.8%
Are penicillins the antibiotics you use most often?			
I agree	283	93.3%	90.03–95.69%
I partly agree	12	4%	2.28–6.79%
I disagree	8	2.7%	1.34–5.12%
Are HPCIA the antibiotics you rarely use?			
I agree	259	85.6%	81.07–89%
I partly agree	26	8.6%	5.92–12.28%
I disagree	18	5.8%	3.79–9.19%
Do you use HPCIA less frequently since the 2018 amendment to the TÄHAV?			
I agree	240	79.2%	74.29–83.4%
I partly agree	31	10.2%	7.3–14.16%
I disagree	32	41.6%	7.58–14.53%
Did the 2018 amendment to the TÄHAV lead to a general reduction in antimicrobial treatments?			
I agree	108	35.6%	30.46–41.19%
I partly agree	69	22.8%	18.41–27.82%
I disagree	126	41.6%	36.17–47.21%
Have you had more AST done since the introduction of the TÄHAV amendment?			
I agree	190	62.7%	57.13–67.96%
I partly agree	60	19.8%	15.71–24.66%
I disagree	53	17.5%	14–22%
Do you generally use penicillins as empirical treatment, due to the TÄHAV amendment?			
I agree	230	75.9%	70.79–80.38%
I partly agree	37	12.2%	8.99–16.38%
I disagree	36	11.9%	8.71–16.01%
After AST, do penicillins more frequently need to be replaced by another substance, compared to other antimicrobial agents?			
I agree	73	24.1%	19.62–29.21%
I partly agree	103	34%	28.89–39.5%
I disagree	127	41.9%	36.49–47.54%
Have you frequently used AST alongside with treatment or only when a change of antibiotic is needed?			
Parallel with treatment	135	44.7%	39.2–50.34%
To change an antibiotic	62	20.5%	16.36–25.44%
Equally often parallel and to change the agent	105	34.8%	29.62–40.3%
For the treatment of which diseases do you need AST particularly often?			
Otitis externa	190	62.7%	57.13–67.96%
Cystitis	168	55.4%	49.82–60.94%
Wounds	132	43.6%	38.1–49.19%
Pyoderma	88	29%	24.22–34.39%
Respiratory Infections	71	23.4%	19.01–28.52%
Diarrhea	47	15.5%	11.87–20.02%
From which species are samples most frequently sent in for resistance testing?			
Cats	19	6.3%	4.05–9.59%
Dogs	116	38.3%	32.99–43.87%
Cats and Dogs equally often	168	55.4%	49.82–60.94%
What are owners’ reactions to additional costs due to antibiogram requirements?			
They agree with submitting a sample.	159	52.5%	46.86–58.03%
They partly agree with submitting a sample.	107	35.3%	30.15–40.85%
They disagree with submitting a sample.	37	12.2%	8.99–16.38%

**Table 2 antibiotics-11-00484-t002:** Antibiotic use and AST. Number, percentages, and 95% CI of responses regarding the diseases of otitis externa, pyoderma, bite wounds, and cystitis according to the frequency of antibiotic use and AST. For this evaluation, 300 questionnaires for otitis externa, 292 for pyoderma, 290 for bite wounds, and 289 for cystitis were analyzed from a survey among German veterinarians.

		Cystitis			Otitis Externa			Pyoderma			Bite Wounds		
		Absolute Number	Percentage	95% CI	Absolute Number	Percentage	95% CI	Absolute Number	Percentage	95% CI	Absolute Number	Percentage	95% CI
Antibiotic use	Always (80–100%)	53	18.30%	14.3–23.21%	95	31.70%	26.66–37.13%	68	23.40%	18.81–28.46%	208	71.70%	66.28–76.6%
	Frequent (60–79%)	95	32.90%	27.71–38.48%	111	37%	31.73–42.6%	89	30.60%	25.48–35.98%	59	20.30%	16.11–25.35%
	Partly (40–59%)	74	25.60%	20.92–30.93%	56	18.70%	14.66–23.46%	71	24.40%	19.75–29.55%	16	5.50%	3.42–8.77%
	Rarely to never (0–39%)	67	23.20%	18.69–28.38%	38	12.70%	9.37–16.91%	63	21.60%	17.24–26.65%	7	2.40%	1.17–4.9%
AST	Always (80–100%)	36	12.50%	9.14–16.76%	18	6%	3.83–9.28%	18	6.20%	3.93–9.53%	13	4.50%	2.64–7.52%
	Frequent (60–79%)	41	14.20%	10.63–18.68%	21	7%	4.62–10.46%	21	7.20%	4.75–10.74%	11	3.80%	2.13–6.66%
	Partly (40–59%)	66	22.80%	18.37–28.01%	42	14%	10.53–18.38%	30	10.30%	7.29–14.29%	28	9.70%	6.76–13.6%
	Rarely (20–39%)	105	36.30%	31–42.02%	147	49%	43.39–54.63%	150	51.40%	45.66–57.05%	95	32.80%	27.61–38.36%
	Never (0–19%)	41	14.20%	10.63–18.68%	72	24%	19.52–29.14%	73	25%	20.38–30.27%	143	49.30%	43.6–55.04%

**Table 3 antibiotics-11-00484-t003:** Cystitis. Number, percentages, and 95% CI of responses regarding the disease of cystitis according to the frequency of the used active ingredients and the antibiotic groups for which a loss of therapeutic success was observed. A total of 289 questionnaires from a survey among German veterinarians were analyzed for this evaluation.

	Absolute Number	Percentage	95% CI
Active ingredient			
Penicillins	216	74.7%	69.43–79.4%
Fluoroquinolones	51	17.6%	13.68–22.46%
Trimethoprim-Sulfonamide	18	6.2%	3.98–9.63%
Others	16	5.5%	3.44–8.8%
Loss of antibiotic therapy success			
No	219	75.8%	70.52–80.36%
Yes	70	24.2%	19.64–29.48%
Penicillins	58/70	82.9%	72.38–89.91%
Fluoroquinolones	11/70	15.7%	9.01–25.99%
Trimethoprim-Sulfonamide	10/70	14.3%	7.95–24.34%

**Table 4 antibiotics-11-00484-t004:** Otitis externa. Number, percentages, and 95% CI of responses regarding the disease of otitis externa according to the frequency of the used active ingredients and the antibiotic groups for which a loss of therapeutic success was observed. A total of 300 questionnaires from a survey among German veterinarians were analyzed for this evaluation.

	Absolute Number	Percentage	95% CI
Active ingredient			
Polymyxin B	135	45%	39.47–50.66%
Gentamicin	85	28.3%	23.53–33.68%
Florfenicol	63	21%	16.77–25.96%
Others	17	5.7%	3.57–8.89%
Loss of antibiotic therapy success			
No	240	80%	75.11–84.13%
Yes	60	20%	15.87–24.89%
Polymyxin B	42/60	70%	57.49–80.1%
Gentamicin	14/60	23.3%	14.44–35.44%
Marbofloxacin	9/60	15%	8.1–26.11%
Florfenicol	4/60	6.7%	2.62–15.93%

**Table 5 antibiotics-11-00484-t005:** Pyoderma. Number, percentages, and 95% CI of responses regarding the disease of pyoderma according to the frequency of the used active ingredients and the antibiotic groups for which a loss of therapeutic success was observed. A total of 292 questionnaires from a survey among German veterinarians were analyzed for this evaluation.

	Absolute Number	Percentage	95% CI
Active ingredient for superficial pyoderma			
Fusidic acid	103	35.4%	30.02–40.91%
Polymyxin B	64	22%	17.55–27.01%
Neomycin	56	19.2%	15.07–24.08%
Active ingredient for deep pyoderma			
Penicillins	164	56.3%	50.43–61.74%
Cephalosporins	113	38.7%	33.29–44.4%
Loss of antibiotic therapy success			
No	253	86.6%	82.26–90.07%
Yes	39	13.4%	9.93–17.74%
Penicillins	34/39	87.2%	73.29–94.4%
Cephalosporins	10/39	25.6%	14.57–41.08%
Fluoroquinolones	3/39	7.7%	2.65–20.32%

**Table 6 antibiotics-11-00484-t006:** Bite wounds. Number, percentages, and 95% CI of responses regarding bite wounds according to the frequency of the used active ingredients and the antibiotic groups for which a loss of therapeutic success was observed. A total of 290 questionnaires from a survey among German veterinarians were analyzed for this evaluation.

	Absolute Number	Percentage	95% CI
Active ingredient			
Penicillins	274	94.5%	91.23–96.58%
Others	16	5.5%	3.42–8.77%
Loss of antibiotic therapy success			
No	263	86.6%	86.79–93.52%
Yes	27	13.4%	6.48–13.21%
Penicillins	23/27	85.2%	67.52–94.08%
Cefovecin	6/27	22.2%	10.61–40.76%
Fluoroquinolones	4/27	14.8%	5.92–32.48%

## Data Availability

The datasets used and/or analyzed during the current study are available from the corresponding author on reasonable request.

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
