# Peer review of "A Cross-Sectional Study of Veterinarians in Germany on the Impact of the TÄHAV Amendment 2018 on Antimicrobial Use and Development of Antimicrobial Resistance in Dogs and Cats"

_antibiotics, 2022, doi:10.3390/antibiotics11040484_

Round 1

Reviewer 1 Report

Thank you for the opportunity to review this interesting manuscript by Moerer et al. entitled "A cross-sectional study of veterinarians in Germany on the impact of the TÄHAV amendment 2018 on antimicrobial use and development of antimicrobial resistance in dogs and cats".

I found the manuscript easy to read. In the introduction, the authors provide an overview of antibiotic resistance and veterinary pharmacy regulations in Germany. The authors clearly report the results obtained from administering a questionnaire to veterinarians in Germany on the use of antibiotics. Although the data collected are subjective opinions of a small proportion of veterinarians, I believe that such an approach is useful to evaluate how these regulations are applied. It will also be interesting to see the data and conclusions from a possible follow-up study.

Author Response

Comments and Suggestions for Authors

Thank you for the opportunity to review this interesting manuscript by Moerer et al. entitled "A cross-sectional study of veterinarians in Germany on the impact of the TÄHAV amendment 2018 on antimicrobial use and development of antimicrobial resistance in dogs and cats".

I found the manuscript easy to read. In the introduction, the authors provide an overview of antibiotic resistance and veterinary pharmacy regulations in Germany. The authors clearly report the results obtained from administering a questionnaire to veterinarians in Germany on the use of antibiotics. Although the data collected are subjective opinions of a small proportion of veterinarians, I believe that such an approach is useful to evaluate how these regulations are applied. It will also be interesting to see the data and conclusions from a possible follow-up study.

Response:

Thank you very much for this very encouraging review.

Reviewer 2 Report

The paper has investigated very important information on the use of antibiotics in veterinary practice in Germany. The data has been compiled with great efforts and deep investigation. I think that this paper must be accepted for publication. 

Author Response

Comments and Suggestions for Authors

The paper has investigated very important information on the use of antibiotics in veterinary practice in Germany. The data has been compiled with great efforts and deep investigation. I think that this paper must be accepted for publication. 

Response:

Thank you very much for this very positive review.

Reviewer 3 Report

The manuscript by Moerer et al. is undoubtedly original, well designed and conducted, and and provides interesting and valuable information on the results of German legislative measures to limit the misuse of antimicrobials and thus against the spread of antibiotic resistance in humans and animals.

I support its publication after appropriate minor modifications as outlined below.

In Materials and Methods section:

  • The authors need to justify their choice of number of 3% of veterinarians for companion animals in the study. In this regard, the authors need to refer to a statistical model, based on which they can validate the survey results, pointed out the previously estimated prevalence. So, the authors must convince the scientific community that they results are completely supportable by statistical tools.

In Discussion section:

  • The authors should mention the additional average costs associated with AST (minimum and maximum percentage limits on treatment costs) – line 218.

The Conclusion section:

  • Please completely rewrite this section focusing only on a succinctly presentation of conclusions derived from the obtained results, as the ideas expressed in the first two phrases (lines 328-332) cannot be considered as conclusions of this study.

Author Response

Comments and Suggestions for Authors

The manuscript by Moerer et al. is undoubtedly original, well designed and conducted, and and provides interesting and valuable information on the results of German legislative measures to limit the misuse of antimicrobials and thus against the spread of antibiotic resistance in humans and animals.

I support its publication after appropriate minor modifications as outlined below.

In Materials and Methods section:

  • The authors need to justify their choice of number of 3% of veterinarians for companion animals in the study. In this regard, the authors need to refer to a statistical model, based on which they can validate the survey results, pointed out the previously estimated prevalence. So, the authors must convince the scientific community that they results are completely supportable by statistical tools.

Response:

This questionnaire was developed from a previous Berlin-wide survey in which there was a high level of response consensus, therefore during the study design the π-value was increased. Thus, with a confidence interval of 95% and a margin of error of 5%, the sample size was 301 participants (l. 306-309). Since the sample of participants is similar to the population of small animal practitioners in Germany in its composition and structure of relevant characteristics, the results can be considered fairly representative (l.288-290).

In Discussion section:

  • The authors should mention the additional average costs associated with AST (minimum and maximum percentage limits on treatment costs) – line 218.

Response:

In consultation with 2 of the most commonly used laboratories in Germany, an antibiogram generates additional costs of 60€ - 70€ for the pet owner (l. 233). However, the laboratories only list the prices for the veterinarians, the veterinarian is obliged to charge an additional fee according to the scale of fees (GOT), the veterinarian is also free to charge additional costs. Therefore, there may be deviations from the stated sum.  

The Conclusion section:

  • Please completely rewrite this section focusing only on a succinctly presentation of conclusions derived from the obtained results, as the ideas expressed in the first two phrases (lines 328-332) cannot be considered as conclusions of this study.

Response:

The conclusion was rewritten and only results of the current study are included now. The first two phrases are deleted, since they did not express actual results of the study.

Reviewer 4 Report

A cross-sectional study of veterinarians in Germany on the impact of the TÄHAV amendment 2018 on antimicrobial use and development of antimicrobial resistance in dogs and cats (antibiotics-1653775).

The manuscript describes the impact of policy changes on the use of antibiotics for the treatment of dogs and cats. It is an important issue to combat antimicrobial resistance but the manuscript needs extensive editing.

The manuscript can be published but following issues should be improved.

Abstract: marginal

Introduction: Very brief and research questions not clear. Please write problem statement and objectives clearly.

Results: L70 3% is not significant from a given population.

L83: Figure 1 denotes under 10% to 81-100%. Please clear x-axis and y-axis of figure 1.

L84 p = 0,403, L92 p = 0.087 please incorporate the p values to the respective tables for every instances and cite Table number in the text to the exact point in results and discussion.

L110-112, 135, 139, 142- data not included in any of the table or difficult to understand. Please make the representation of the result easier by citing table number in this lines and all other instances.

Tables 2-5 should incorporate in only one Table.

Discussion: marginal. Should be based on findings. This work has many limitations.

Materials and methods: the main findings of all statistical analyses should incorporate in the manuscript or a supplementary file.

Conclusions: L334 should present data statistically

English correction: I am not a native speaker but I think the manuscript needs English correction for better readability.

L14 requirements should be amendment

L32 selection should be emergence or development

L11 minimize and L33 combatting- mixture of American and British English and many more.

Reviewer 5 Report

A cross-sectional study of veterinarians in Germany on the impact of the TÄHAV amendment 2018 on antimicrobial use and development of antimicrobial resistance in dogs and cats

Thanks for the opportunity to review this paper.

To begin with, I applaud the authors for their hard work and efforts in extensively gathering the data to understand the impact of an antibiotic use intervention in Germany. This study is of remarkable significance to the public health community and is of personal interest.

One of the salient shortcomings of this paper is that it is very descriptive without any specific hypothesis tested despite having set out a decent study design. They set out to test the impact of amendment of the TÄHAV 2018 on antibiotic use especially of HPCIA amongst German veterinarians, but their analysis is just a description of prescription patterns amongst interviewed veterinarians. Authors should change the focus of the paper throughout to reflect what they did.

Authors do not describe how they calculated antibiotic use and instead they estimate daily use of antimicrobials qualitatively. This is misleading.

The finding of lack of association between loss of efficacy of antimicrobial drugs and size of the city seems misplaced and confounded.

Figure 1 lack axes labels

Result section is just an extended description of tables and summary of findings. Authors have attempted to analyze drivers of prescription patterns but have not done so systematically using statistical approaches.

Author Response

Thanks for the opportunity to review this paper.

To begin with, I applaud the authors for their hard work and efforts in extensively gathering the data to understand the impact of an antibiotic use intervention in Germany. This study is of remarkable significance to the public health community and is of personal interest.

One of the salient shortcomings of this paper is that it is very descriptive without any specific hypothesis tested despite having set out a decent study design. They set out to test the impact of amendment of the TÄHAV 2018 on antibiotic use especially of HPCIA amongst German veterinarians, but their analysis is just a description of prescription patterns amongst interviewed veterinarians. Authors should change the focus of the paper throughout to reflect what they did.

The purpose of this study was only to investigate the influence of the TÄHAV amendment on the use of antibiotics and not the general influencing factors.

The manuscript has been revised to clearly show that opinions and views on antibiotic use and AST use among German small animal veterinarians were collected during the survey and these qualitative data are presented in this paper.

Authors do not describe how they calculated antibiotic use and instead they estimate daily use of antimicrobials qualitatively. This is misleading.

In Germany, it is currently not mandatory to record the amount of antibiotics used in the treatment of dogs and cats. Therefore, the participants could only be asked to estimate the percentage of antibiotics used daily. A precise query of the quantities will only be possible from 2026, as the amendment to the Medicines Act will then oblige German veterinarians to report used antimicrobials (l.190-192).

L.84 clearly states that these are only estimates.

The finding of lack of association between loss of efficacy of antimicrobial drugs and size of the city seems misplaced and confounded.

The association between city size and loss of efficacy of antibiotic therapies was removed from the manuscript because these are only subjective assessments and thus cannot be statistically validated.

Figure 1 lack axes labels

The x-axis describes the estimated percentage of antibiotic treatments per day per practice and the y-axis shows the absolute number of participants. The axis titles have been added to the graph.

Result section is just an extended description of tables and summary of findings. Authors have attempted to analyze drivers of prescription patterns but have not done so systematically using statistical approaches.

All questions that related to one disease were analysed in multivariable logistic regression models. We did not want to mix variables of different questions to avoid confounding. Thus, we ended up in a series of relatively small models.

It must be noted that all results are based on the subjective views and opinions of the participants and cannot be supported by objectively generated data. Therefore, statistical analysis of these data must also be done cautiously, as the data obtained reflect perceived tendencies rather than facts. Therefore, we decided to work primarily descriptively.

Round 2

Reviewer 5 Report

Manuscript quality has improved greatly.

Author Response

Thank you very much for your support of our study!